# IPMark: A Sentence-Level Watermark for LLMs with Hierarchical Personalization and Efficient Detection

Wenbo An [1]   Lianwei Wu [* 1 2]   Zehao Wang [1]

## Abstract

Watermarking has emerged as a critical solution for the detection and provenance tracing of content generated by large language models. However, existing methods still suffer from significant limitations, including difficulties in achieving efficient and personalized attribution, substantial degradation of generation quality, and low robustness against attacks. To address these challenges, we propose IPMark, the first IP-inspired hierarchical personalized watermarking framework. Specifically, to enable personalization and efficient detection, IPMark employs a hierarchical addressing framework to structurally organize model and user identities. Subsequently, addressing the inherent semantic distortion caused by token-level watermarking, we design a semantic-syntactic dual-stream embedding strategy. Centered on sentence-level candidate selection and reinforced by dual signals from syntactic and semantic features, this approach optimizes the injection process, thereby significantly enhancing generation quality while ensuring strong robustness. Experimental results demonstrate that IPMark achieves the lowest perplexity among baselines, ensuring superior generation quality while maintaining strong robustness and significantly reducing detection latency through hierarchical retrieval. Our code is available at https://github.com/nwlt/IPMark.

## 1. Introduction

The emergence of Large Language Models (LLMs) (Zhang et al., 2022) has transformed text-processing tasks such as poetry composition, document translation, and press-release drafting. However, their widespread use has also raised a range of concerns, including copyright infringement, the proliferation of low-quality content, and the spread of misinformation. As LLMs are deployed more broadly, it is becoming increasingly important to accurately trace LLM-generated content to attribute responsibility to specific users of particular models. To address this challenge, LLM watermarking (Kirchenbauer et al., 2023; Zhao et al., 2023; Kuditipudi et al., 2023; Zhao et al., 2025; Wang et al., 2025) has emerged as a pivotal solution that implants statistically detectable and imperceptible structural patterns into generated text, thereby establishing provenance while preserving semantic fidelity.

Text watermarking techniques are categorized into inference-time and output-time approaches based on the insertion stage (Kirchenbauer et al., 2023; Chen et al., 2025; Liu et al., 2023; Niess & Kern, 2025; Mao et al., 2025; Lin et al., 2025; Zhang et al., 2025a; Yu et al., 2026; Chang et al., 2024; Lau et al., 2024). Inference-time watermarking dynamically intervenes in the decoding process to imprint patterns, comprising token-level methods that introduce dedicated sampling rules for individual tokens and sentence-level methods that adjust syntactic relations or sentence structures. In contrast, output-time watermarking functions as a rule-based post-processing mechanism that embeds signals by editing text after generation—ranging from lightweight rule modifications to LLM-based rewriting—without requiring access to the model's parameters.

There are two major challenges in deploying watermarking in practice. **First**, current research predominantly focuses on model-level provenance, often neglecting the efficient detection of personalized watermarks. This lack of efficient user-level attribution prevents the precise identification of individual users, which is essential for real-world accountability (Yoo et al., 2023). **Second**, because most watermarking schemes embed signals by modulating token logits to induce statistically detectable patterns, the resulting text can suffer a substantial quality degradation relative to the unwatermarked model (Huang et al., 2025; Giboulot & Furon, 2024), and the watermark can be readily removed by adversaries through simple edits (Liang et al., 2025; Wu & Chandrasekaran, 2024; Zhang et al., 2025b) such as replace-

---

[1]Northwestern Polytechnical University, Xi'an, Shaanxi, China [2]Shenzhen Research Institute of Northwestern Polytechnical University, Shenzhen, Guangdong, China. Correspondence to: Lianwei Wu <wlw@nwpu.edu.cn>.

*Proceedings of the 43rd International Conference on Machine Learning*, Seoul, South Korea. PMLR 306, 2026. Copyright 2026 by the author(s).

ment, back-translation, or paraphrasing.

Inspired by the hierarchical addressing scheme of IP addresses in computer networks—where the network part and host part are encoded separately—we propose IPMark, a watermarking method that mimics the IP-address format to achieve hierarchical embedding for both model and user identities, thereby enabling personalization and efficient detection. Specifically, to facilitate efficient detection, we establish a coarse-to-fine verification framework that emulates the filtering-and-forwarding logic of network routing. This module classifies watermark units based on discriminative semantic features and progressively validates the source via a two-stage lookup: first identifying the model (network segment) through a Model Library match, and subsequently pinpointing the specific user (host address) via a User Library query.

Correspondingly, to improve generation quality while maintaining strong robustness, we develop a hierarchical embedding mechanism driven by semantic-syntactic synergy. By extracting semantic triples to retrieve hypernyms (Miller, 1995) from an open knowledge graph as core signals—and supplementing them with syntactic dependency cues—this mechanism integrates these structural features with sentence-level log-probabilities to strictly optimize candidate sentence selection, thereby embedding identity information without disrupting the textual flow.

Extensive evaluations on OPT-1.3B (Zhang et al., 2022) and Llama-3 (Grattafiori et al., 2024) demonstrate that IPMark achieves the lowest perplexity (2.779), reflecting a 3.9% quality improvement over the best-performing baseline, reduces detection latency by up to 93% through hierarchical retrieval, and maintains robustness with AUC exceeding 0.99 against lexical perturbations. The main contributions of this paper are summarized as follows:

1. **The first IP-inspired Hierarchical Personalized Watermarking Framework:** Drawing inspiration from the hierarchical addressing principle of computer networks, we design a precise model-user level tracing framework. By structurally decoupling model signatures from user identities, this approach enables accurate and scalable provenance tracking from the foundation model down to individual users.

2. **Optimal Trade-off between Quality and Robustness:** Our method achieves the lowest perplexity of 2.779 among all comparative settings—surpassing even the competitive Unigram baseline by 3.9%—while ensuring superior resilience against complex attacks, including synonym substitution and cross-lingual translation.

3. **High-Efficiency Detection at Scale:** We introduce a hierarchical retrieval strategy that significantly enhances system throughput. Experimental results show

that for a scale of 100,000 users, logical partitioning reduces detection latency to 5.020 seconds, achieving a 14-fold acceleration compared to linear scanning. Furthermore, in large-scale scenarios with 1,000,000 users, our mechanism completes watermark verification in just 5.928 seconds, reducing latency by approximately 93% and ensuring the feasibility of real-time monitoring in expansive industrial environments.

## 2. Related Work

To promote the safe and lawful use of large language models, researchers have proposed embedding watermarks directly into model-generated content. Depending on when the watermark is inserted, text watermarking techniques can be further categorized into inference-time watermarking and output-time watermarking.

**Inference-time Watermarking.** Inference-time watermarking operates by dynamically embedding provenance signals into the output of Large Language Models (LLMs) during the stochastic generation process. Technically, this involves intervening in the sampling logic—modulating the raw logit scores of candidate tokens—to imprint identifiable statistical patterns without disrupting semantic coherence.

Depending on the embedding granularity, existing approaches are categorized into token-level and sentence-level strategies. Token-level methods (Kirchenbauer et al., 2023; Chen et al., 2025; Liu et al., 2023; Niess & Kern, 2025; Qu et al., 2025; Mao et al., 2025) introduce biased sampling rules at each generation step. While computationally efficient, they often perturb the optimal language distribution, leading to detectable degradation in text quality. Conversely, sentence-level watermarking (Dabiriaghdam & Wang, 2025; Lin et al., 2025; Zhang et al., 2025a; Xu et al., 2024) embeds information by manipulating macroscopic structures. Although this preserves coherence better, it typically incurs higher latency and lacks the granularity required for precise user tracing. Consequently, existing inference-time methods struggle to achieve a Pareto optimum between imperceptibility and fine-grained traceability. In this work, we propose a hierarchical watermark-embedding scheme inspired by IP hierarchical addressing in computer networks. Unlike direct sampling interventions, our approach employs personalized candidate sentence filtering, enabling fine-grained, user-level tracing while ensuring high-quality text generation.

**Output-time Watermarking.** Output-time watermarking, also referred to as post-processing watermarking, functions by modifying the generated text after the inference phase is complete. These approaches (Zhang et al., 2024; Yu et al., 2026; Hao et al., 2025) operate as black-box wrappers, obviating the need for access to the model's internal parameters.

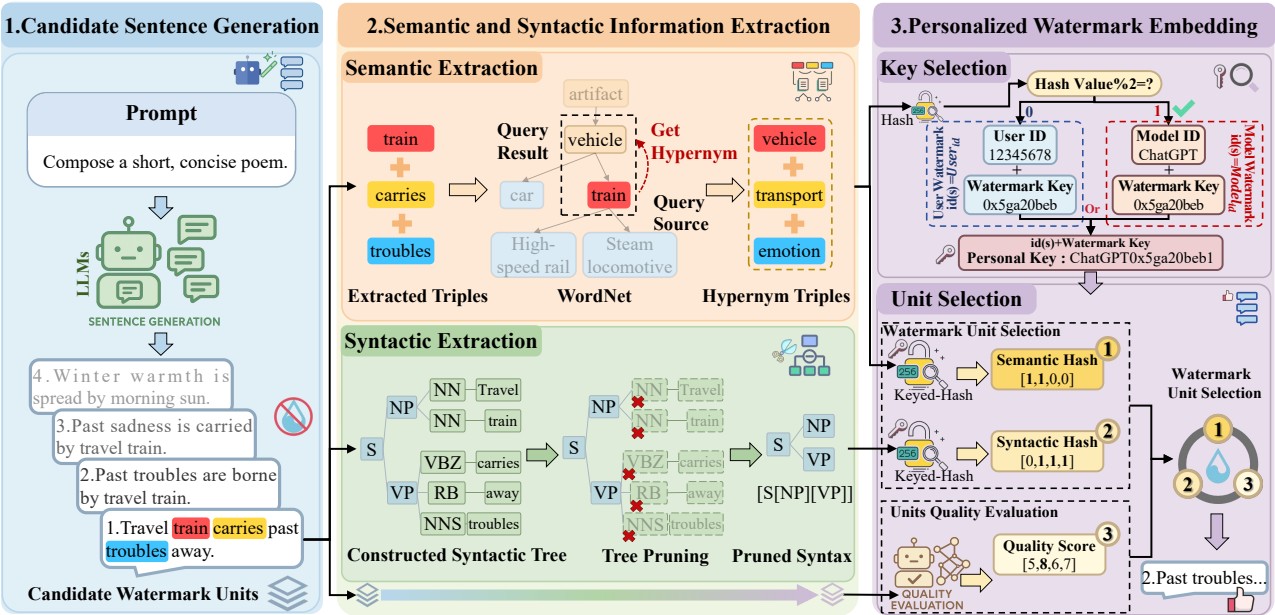

*Figure 1.* Schematic of the hierarchical IPMark personalized watermark-embedding framework. The framework consists of three main stages: (i) Candidate sentence generation: the LLM produces candidate sentences from the user prompt and parses them into basic watermark units; (ii) Information extraction: robust syntactic and semantic features are extracted from the watermark units by jointly leveraging semantic triple extraction and constituency parsing; and (iii) Personalized watermark embedding: a personalized watermark key is derived by combining the user secret key with the model identifier, and the optimal unit is selected—based on semantic and syntactic hashing as well as quality assessment—to complete watermark insertion. This design jointly ensures high capacity, strong robustness, and high-quality text generation.

Depending on the specific embedding mechanism, existing methodologies can be categorized into: (i) lightweight rule-based modifications (Sato et al., 2023); (ii) statistical feature injection via rejection sampling (Dabiriaghdam & Wang, 2025; Yu et al., 2026); (iii) semantic fusion through LLM-based rewriting (Chang et al., 2024; Lau et al., 2024); and (iv) specialized post-processing for code generation (Yang et al., 2024). While these paradigms have advanced specific performance dimensions, they face a significant bottleneck in balancing the 'impossible triangle' of watermarking. Lightweight methods typically suffer from limited capacity and vulnerability to adversarial edits, whereas heavy-weight semantic rewriting often incurs significant computational overhead or inadvertently distorts the original semantic intent.

To overcome these inherent limitations, we propose a hierarchical watermark-embedding scheme inspired by IP hierarchical addressing in computer networks. Unlike fragile rule-based post-processing, our framework leverages structured semantic decoupling to achieve fine-grained, user-level tracing. This approach effectively resolves the robustness-fidelity trade-offs found in prior work, ensuring robustness against attacks while maintaining the high fidelity of the original text generation.

# 3. Hierarchical Watermark Embedding

In this section, we propose IPMark, a hierarchical personalized watermarking framework inspired by the hierarchical addressing scheme of IP networks. As illustrated in Figure 4, IPMark achieves hierarchical and efficient provenance tracking at both the model and user levels for large language models, effectively enhancing watermark robustness while minimizing the impact on generation quality. Specifically, IPMark comprises three integral modules: Candidate Sentence Generation, Semantic and Syntactic Information Extraction, and Personalized Watermark Embedding. We provide a detailed exposition of each module in the following subsections.

## 3.1. Candidate Sentence Generation

We first construct a set of candidate sentences at each generation step to provide sufficient flexibility for subsequent watermark embedding without altering task semantics. Specifically, under the same contextual condition $c$, the LLM produces multiple candidates that are semantically similar yet exhibit slight variations in wording and phrasing (Giboulot & Furon, 2024). Formally, we obtain a candidate set containing $K$ sentences to be embedded with watermark:

$$\mathcal{C}(c) = \{s_1, \ldots, s_K\}, \quad s_i \sim p_\theta(\cdot \mid c). \tag{1}$$

Here, $p_\theta(\cdot \mid c)$ denotes the LLM distribution conditioned on the context $c$. This expansion of the solution space lays the foundation for hierarchical watermarking. Because IPMark must support both model-level attribution and user-level tracing, the candidate set provides increased degrees of freedom, allowing the mechanism to accommodate selection preferences induced by different keys. By exploring this plausible space, we minimize the need to deviate from the model's natural tendencies, thereby preserving fluency and readability.

### 3.2. Syntactic and Semantic Information Extraction

To ensure the watermark is robust against perturbations while maintaining high discriminability, we extract both semantic and syntactic features from each candidate $s$.

**Semantic Extraction.** We first extract semantic triples (Dozat & Manning, 2016) (subject–relation–object) and map key concepts to abstract hypernyms using an open knowledge graph (Miller, 1995). Unlike surface-level lexical forms, hypernyms are resilient to synonym substitution and local rewriting. We denote this semantic signature by:

$$\mathcal{H}(s) = \phi(\tau^*), \quad \tau^* = \arg\max_{\tau \in \mathcal{T}(s)} \text{Salience}(\tau), \quad (2)$$

where $\mathcal{T}(s)$ is the set of extracted candidate triples, $\text{Salience}(\cdot)$ evaluates the semantic importance (e.g., based on the dependency parse tree depth) to identify the primary triple $\tau^*$, and $\phi(\cdot)$ maps its concepts to stable hypernyms.

**Syntactic Extraction.** Relying solely on semantic cues may be insufficient when candidates are semantically very close, as the decision signal becomes weak. To provide complementary discriminative signals, we extract the constituency parse tree $\text{CTree}(s)$ and apply a pruning operator $\pi(\cdot)$ to remove redundant subtrees that do not affect the main meaning (Dozat & Manning, 2016). The pruned tree is then linearized into a string representation:

$$\begin{aligned}
T = \text{CTree}(s) &= \big(\ell, \langle\tau_1, \ldots, \tau_m\rangle\big), \\
\mathcal{S}(s) &= \text{Lin}(\pi(T)), \quad \text{where} \\
\text{Lin}(T) &= \text{"("} \parallel \ell \parallel \text{" "} \parallel \Big( \big\|_{j=1}^{m} \text{Lin}(\tau_j) \Big) \parallel \text{")"}.
\end{aligned} \quad (3)$$

Here, we formalize the parse tree $T$ as a tuple containing a constituent label $\ell$ (e.g., NP, VP) and a sequence of subtrees $\langle\tau_j\rangle$. The linearization function $\text{Lin}(\cdot)$ is defined recursively, where $\parallel$ denotes string concatenation and the large operator signifies the sequential concatenation of the linearized subtrees, transforming the pruned hierarchical structure into a canonical bracketed string. In this framework, semantic cues $\mathcal{H}(s)$ provide stability, while syntactic cues $\mathcal{S}(s)$ improve discriminability.

### 3.3. Personalized Watermark Embedding

We employ a hierarchical embedding strategy that preserves both model-level and user-level traceability.

**Key Selection.** For each candidate $s$, we determine whether to embed the user ID or the model ID based on its semantic signature:

$$\begin{aligned}
z(s) &= \text{Hash}(\mathcal{H}(s)) \bmod 2, \\
\text{id}(s) &= \begin{cases} \text{user\_id}, & z(s) = 0, \\ \text{model\_id}, & z(s) = 1. \end{cases}
\end{aligned} \quad (4)$$

Here, $z(s) \in \{0, 1\}$ serves as the layer selector and $\text{Hash}(\cdot)$ is a cryptographic hash.

**HMAC Computation.** We derive a sentence-specific key by concatenating a base secret $k_0$ with the selected identifier, and compute hash values using the Hash-based Message Authentication Code (HMAC) (Krawczyk et al., 1997) for both semantic and syntactic representations:

$$\begin{aligned}
k(s) &= \text{Concat}(k_0, \text{id}(s)), \\
\mathcal{M}_{k^*(s)}(m) &= \mathsf{H}\Big( (k^*(s) \oplus \text{opad}) \parallel \\
&\qquad \mathsf{H}\big((k^*(s) \oplus \text{ipad}) \parallel m\big)\Big), \\
h_{\text{sem}}(s) &= \mathcal{M}_{k^*(s)}(\mathcal{H}(s)), \\
h_{\text{syn}}(s) &= \mathcal{M}_{k^*(s)}(\mathcal{S}(s)).
\end{aligned} \quad (5)$$

Here, $\mathsf{H}(\cdot)$ is the underlying hash function (e.g., SHA-256), $\oplus$ denotes bitwise XOR, $\parallel$ denotes concatenation, $\text{ipad}$ and $\text{opad}$ are standard padding constants, and $k^*(s)$ denotes $k(s)$ after hashing (if needed) and padding/truncation to the hash block size.

**Unit Selection.** Finally, we select the best candidate $s^\star$ by prioritizing watermark feasibility (passing the HMAC-based checks) and then optimizing generation quality (maximizing log-probability). We define feasible sets based on whether the HMAC outputs map to the target residue class (e.g., $\equiv 1 \pmod 2$):

$$\begin{aligned}
\mathcal{F}_{\text{both}} &= \Big\{ s \in \mathcal{C}(c) \Big| h_{\text{sem}}(s) \equiv h_{\text{syn}}(s) \equiv 1 \pmod 2 \Big\}, \\
\mathcal{F}_{\text{sem}} &= \Big\{ s \in \mathcal{C}(c) \Big| h_{\text{sem}}(s) \equiv 1 \pmod 2 \Big\}, \\
\mathcal{F}_{\text{syn}} &= \Big\{ s \in \mathcal{C}(c) \Big| h_{\text{syn}}(s) \equiv 1 \pmod 2 \Big\},
\end{aligned} \quad (6)$$

$$s^\star = \begin{cases} \arg\max_{s \in \mathcal{F}_{\text{both}}} \log p_\theta(s \mid c), & \mathcal{F}_{\text{both}} \neq \varnothing, \\ \arg\max_{s \in \mathcal{F}_{\text{sem}}} \log p_\theta(s \mid c), & \mathcal{F}_{\text{sem}} \neq \varnothing, \\ \arg\max_{s \in \mathcal{F}_{\text{syn}}} \log p_\theta(s \mid c), & \mathcal{F}_{\text{syn}} \neq \varnothing, \\ \arg\max_{s \in \mathcal{C}(c)} \log p_\theta(s \mid c), & \text{otherwise.} \end{cases}$$

Here, $\mathcal{F}_{\text{both}}$, $\mathcal{F}_{\text{sem}}$, and $\mathcal{F}_{\text{syn}}$ are feasible subsets prioritized in descending order, and $\varnothing$ denotes the empty set. When all feasible subsets are empty, we use $\mathcal{C}(c)$ as a default fallback set. This selection rule prioritizes candidates that pass both semantic and syntactic HMAC checks; if such candidates are unavailable, it falls back to those passing the semantic check, then to those passing the syntactic check, and finally to the full candidate set $\mathcal{C}(c)$ as a last resort. Within the selected set, we choose the candidate with the highest sentence-level log-probability, thereby favoring watermark-feasible outputs while preserving text quality.

## 4. Hierarchical Watermark Detection

In this section, we present a hierarchical detection framework for IPMark. We consider the following null hypothesis $H_0$ and the alternative $H_1$:

$H_0$ : The candidate text is not watermarked.

$H_1$ : The candidate text is watermarked.

Given a candidate text $X$, we first parse it into a sequence of discrete sentences $S = \{s_1, \ldots, s_N\}$. Based on the semantic signatures, we partition these sentences into two sets: the model-identifying sentences $S_M$ and the user-identifying sentences $S_U$. Recalling the layer selector $z(\cdot)$ defined in Eq. (4), this partition is formally given by:

$$S_M = \big\{ s \in S \mid z(s) = 1 \big\},$$
$$S_U = \big\{ s \in S \mid z(s) = 0 \big\}. \tag{7}$$

The detection proceeds in a coarse-to-fine manner. We define a scoring function $\text{Score}(S, \text{id})$ that counts the number of sentences in a set $S$ satisfying the watermark constraints under a candidate identity id. Let $h_{\text{sem}}(s)$ and $h_{\text{syn}}(s)$ denote the HMAC outputs computed as per Eq. (5) using the key derived from id (i.e., $k = \text{Concat}(k_0, \text{id})$). The score is defined as:

$$\text{Score}(S, \text{id}) = \sum_{s \in S} \Big[ \delta \cdot \big( h_{\text{sem}}(s) \bmod 2 \big)$$
$$+ (1 - \delta) \cdot \big( h_{\text{syn}}(s) \bmod 2 \big) \Big]. \tag{8}$$

The parameter $\delta \in [0, 1]$ is a coefficient that regulates the trade-off between semantic and syntactic information in the watermark detection process. Specifically, $\delta$ scales the contribution of the semantic-based hash ($h_{\text{sem}}$), while ($1 - \delta$) scales the syntactic-based hash ($h_{\text{syn}}$). By tuning this parameter, the scoring function can be adaptively optimized: a higher $\delta$ emphasizes the preservation of meaning, whereas a lower $\delta$ prioritizes the structural integrity of the text.

We start from the model level using a pre-built model repository $\mathcal{R}_{\text{model}}$. We identify the candidate model $\hat{m}$ by maxi-

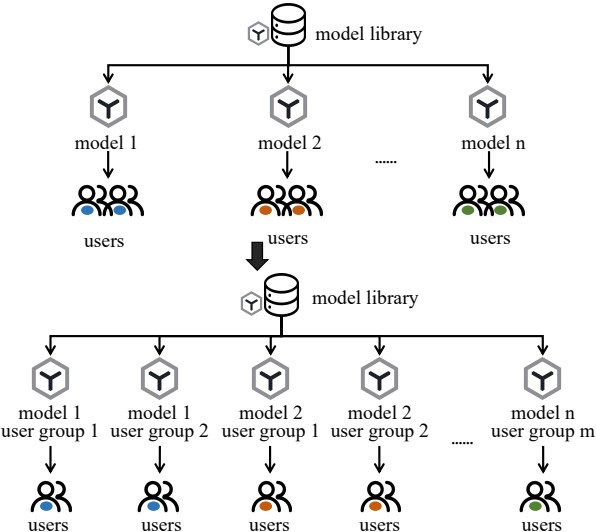

*Figure 2.* The transformation of the model library structure from a conventional one-to-one mapping to a hierarchical model-user group partitioning mechanism.

mizing the matching score on $S_M$:

$$\hat{m} = \arg \max_{m \in \mathcal{R}_{\text{model}}} \text{Score}(S_M, m). \tag{9}$$

If $\text{Score}(S_M, \hat{m}) \leq \tau_M$ (where $\tau_M$ is a significance threshold), we accept $H_0$. If the model-level matching succeeds, we proceed to the user level using the user-ID repository $\mathcal{R}_{\text{user}}(\hat{m})$ associated with $\hat{m}$. We identify the concrete user identity $\hat{u}$ by maximizing the score on $S_U$:

$$\hat{u} = \arg \max_{u \in \mathcal{R}_{\text{user}}(\hat{m})} \text{Score}(S_U, u). \tag{10}$$

If $\text{Score}(S_U, \hat{u}) \leq \tau_U$, we accept $H_0$. Only when both stages succeed do we accept $H_1$ and output the joint attribution result $(\hat{m}, \hat{u})$. This two-level validation structures detection as model identification followed by user localization, providing a robust basis for accountability.

To optimize efficiency, we leverage a model-user hierarchical embedding mechanism, as illustrated in Figure 2. Specifically, we refine the model library from a conventional 'one model corresponds to one user database' structure into granular logical blocks, such as 'Model 1–User Group 1', 'Model 1–User Group 2', and 'Model 2–User Group 1'. This hierarchical partitioning aligns the user repository with the model structure, allowing the detection system to narrow down the search space by first identifying the model category and subsequently locating the specific user group. This coarse-to-fine strategy significantly reduces search complexity compared to flat repository structures, facilitating highly efficient hierarchical retrieval.

*Table 1.* Performance comparison of our method and baselines under various attack scenarios.

| Method | No Attack | | Synonym-5 | | Synonym-10 | | Synonym-15 | | Copy-Paste-50 | | Dipper Rewrite | | Translation | |
|---|---|---|---|---|---|---|---|---|---|---|---|---|---|---|
| | AUC | F1 | AUC | F1 | AUC | F1 | AUC | F1 | AUC | F1 | AUC | F1 | AUC | F1 |
| KGW | 0.997 | 0.995 | 0.997 | 0.991 | 0.996 | 0.989 | 0.994 | 0.989 | 0.950 | 0.898 | 0.764 | 0.713 | 0.835 | 0.760 |
| Unigram | 0.992 | 0.987 | 0.983 | 0.970 | 0.983 | 0.968 | 0.983 | 0.967 | 0.827 | 0.768 | 0.587 | 0.666 | 0.821 | 0.758 |
| EXP | 0.997 | **0.997** | 0.994 | **0.995** | 0.995 | **0.994** | 0.995 | **0.994** | **0.983** | **0.972** | 0.732 | 0.692 | 0.815 | 0.759 |
| PF | 0.996 | 0.995 | 0.991 | 0.992 | 0.990 | 0.991 | 0.989 | 0.991 | 0.941 | 0.956 | **0.874** | **0.894** | **0.922** | **0.938** |
| MorphMark | 0.995 | 0.987 | 0.991 | 0.979 | 0.991 | 0.976 | 0.990 | 0.976 | 0.907 | 0.842 | 0.729 | 0.694 | 0.905 | 0.834 |
| **Ours** | **0.999** | 0.985 | **0.998** | 0.974 | **0.997** | 0.975 | **0.995** | 0.966 | 0.871 | 0.804 | 0.773 | 0.749 | 0.840 | 0.765 |

*Table 2.* Comparison of Perplexity (PPL) results.

| Method | Natural | KGW | Unigram | EXP | PF | MorphMark | UWM | **Ours** |
|---|---|---|---|---|---|---|---|---|
| **PPL** | 8.893 | 3.733 | 2.892 | 3.250 | 8.056 | 3.251 | 2.804 | **2.779** |

## 5. Experiments

### 5.1. Experiment Setting

**Implementation Details.** In the main experiments, we evaluate the effectiveness of IPMark on three large language models with different scales and architectures: OPT-1.3B (Zhang et al., 2022), Llama-3.2-3B, and Llama-3.1-8B (Grattafiori et al., 2024). During generation, we implement a stochastic sampling strategy with the temperature hyperparameter set to 0.7 and the top-p value maintained at 0.9. This configuration is chosen to provide sufficient cumulative probability for watermark embedding while preserving the semantic coherence and creative diversity of the model output. We perform sentence-by-sentence text generation based on the C4 dataset (Raffel et al., 2020). For each sample, we use the first 25 tokens as the prompt and iteratively generate until 20 accepted sentences are obtained. In each iteration, we employ beam search to produce a candidate set of size 10. Sentence boundaries are detected using regular-expression rules that identify end-of-sentence punctuation or newline characters, and each completed path is decoded as a candidate sentence. Candidates are scored by cumulative log-probability and ranked with length normalization, using a length-penalty coefficient 0.7. We then select the final sentence from the candidate set according to the watermark priority, and append its tokens to the output sequence. For reproducibility, we fix the random seed in all experiments.

**Baselines.** To comprehensively compare IPMark against prior work, we include multiple representative watermarking methods as baselines, including KGW (Kirchenbauer et al., 2023), Unigram (Zhao et al., 2023), EXP (Kuditipudi et al., 2023), Permute-and-Flip (PF) (Zhao et al., 2025) and MorphMark (Wang et al., 2025).

**Evaluation Metrics.** Under the same model settings, we evaluate each method along three dimensions. (i) Watermark performance is assessed by benchmarking detectability prior to attacks and robustness following adversarial perturbations, utilizing ROC-related metrics including ROC-AUC and Best-F1. (ii) Text quality is measured using perplexity, computed by Llama-3.1-8B. (iii) We report the runtime and computational overhead of watermark embedding and detection to support an overall assessment of practical usability.

### 5.2. Detectability and Robustness Evaluation

**Attack Methods.** To systematically evaluate the robustness of IPMark under realistic adversarial scenarios, we construct a comprehensive set of attack methods spanning three representative categories of perturbations: lexical editing, semantic rewriting, and cross-lingual transformation.

- **Text Editing:** We employ two operations. First, *Synonym Substitution* utilizes a BERT-based replacement model to randomly substitute words with their synonyms at varying rates ($\gamma \in \{5\%, 10\%, 15\%\}$), mimicking mild to moderate revision. Second, *Copy–Paste Mixing* dilutes the watermark signal by concatenating watermarked text with non-watermarked text at a fixed ratio (50%), simulating scenarios where generated content is interspersed with human writing.

- **Automatic Rewriting:** We leverage DIPPER to paraphrase the watermarked text (Krishna et al., 2023), configured with lexical diversity set to 40 and order diversity set to 20. This simulates meaning-preserving attacks where sentence structure and surface realizations are reorganized while retaining original semantics.

- **Cross-Lingual Transformation:** To assess stability under severe distribution shifts, we introduce a *Back-Translation* (He et al., 2024) attack (English → Chinese → English). This process induces significant lexical and syntactic changes through cross-language mapping, serving as a stress test for watermark survivability.

**Results and Analysis.** Table 1 summarizes the detection

performance (AUC-ROC and Best-F1) across various attack settings. The results presented here are based on the Llama-3.2-3B model; for additional results on models such as Llama-3.1-8B and OPT-1.3B, please refer to Appendix A. Overall, while detection capability inevitably degrades as editing intensity increases, our method (IPMark) exhibits superior stability, particularly in AUC-ROC metrics, compared to standard baselines.

In the *No Attack* setting, IPMark achieves near-perfect detection with an AUC of 0.999, outperforming other baselines. Under *Synonym Substitution* attacks, IPMark demonstrates exceptional resilience; even as the substitution rate increases to 15%, our method maintains an AUC above 0.995. This significantly surpasses unigram-based methods and exhibits greater stability than EXP and KGW, suggesting that our semantic-based embedding effectively mitigates the impact of local lexical shifts.

Regarding complex semantic perturbations, IPMark maintains competitive robustness. In the challenging *Cross-Lingual* (En–Zh–En) setting, IPMark achieves an AUC of 0.840 and F1 of 0.765, successfully surpassing widely-used purely distributional watermarks such as KGW (AUC 0.835) and EXP (AUC 0.815). Similarly, under *Dipper Rewrite* attacks, IPMark (AUC 0.773) outperforms both KGW (0.764) and EXP (0.732). Although specific robustness-oriented methods like PF achieve higher scores in these high-entropy tasks, IPMark strikes a balanced performance, offering a resilient attribution mechanism that remains effective against both surface-level edits and deep adversarial semantic restructuring.

## 5.3. Text Quality

Text quality is assessed using Perplexity (PPL) (Berger et al., 1996), where a lower score indicates higher fluency and better alignment with the evaluation model's distribution. We employ `Llama-3.1-8B` as the oracle evaluator to score 1,000 samples generated by `Llama-3.2-3B`.

As summarized in Table 2, our method achieves the lowest PPL of 2.779 across all evaluated watermarking methods. In comparison with watermarking baselines, IPMark outperforms KGW (3.733), EXP (3.250), and MorphMark (3.251), and further improves upon the strong Unigram baseline (2.892). The robustness-focused method PF records a significantly higher PPL of 8.056, which is close to the complexity of Natural Text (8.893) but suggests higher generation entropy.

Crucially, the PPL of IPMark (2.779) is comparable to, and even slightly lower than, that of the baseline unwatermarked model (UWM, 2.804). This result is significant as it demonstrates that our watermarking strategy introduces negligible perturbation to the generation quality. Unlike methods that inadvertently degrade fluency, IPMark effectively preserves the intrinsic statistical characteristics of the original model distribution while ensuring robust detectability.

## 5.4. Traceability Efficiency and Computational Overhead

*Table 3.* Comparison of Average Detection Latency across Different Retrieval Strategies and Library Scales (Averaged over 10 runs).

| User Scale | Retrieval Strategy | Configuration | Avg. Latency (s) |
|---|---|---|---|
| 1 | Linear Scan | Single ID Match | 0.063 |
| 10 | Linear Scan | - | 0.497 |
| 100 | Linear Scan | - | 2.281 |
| 1,000 | Linear Scan | - | 2.964 |
| 10,000 | Linear Scan | - | 8.198 |
| 100,000 | Linear Scan | - | 72.677 |
| **100,000** | **Hierarchical** | **10 × 10,000** | **8.695** |
| **100,000** | **Hierarchical** | **100 × 1,000** | **5.020** |
| **1,000,000** | **Hierarchical** | **1,000 × 1,000** | **5.928** |

The traceability performance of IPMark is intrinsically tied to the predefined repository of Model Identifiers (IDs) and User IDs. To achieve precise attribution of the text under inspection, the system must traverse the candidate sets and execute watermark verification operators or similarity calculations for each entry.

**Resource Initialization and Pre-loading.** Prior to initiating the detection pipeline, critical linguistic resources are loaded into the resident memory to eliminate I/O latency during real-time processing:

- **Syntactic and Semantic Analysis Pipeline**: The system integrates the high-performance spaCy processing model, en_core_web_sm, to perform fine-grained tokenization, Part-of-Speech (POS) tagging, and dependency parsing.

- **Semantic Knowledge Base**: The WordNet lexical semantic data from NLTK is pre-loaded to provide ontological support for verifying watermark signals within the semantic space.

**Detection Performance Evaluation and Hierarchical Retrieval Mechanism.** We evaluate the detection latency across various library scales. To ensure the reliability of the results, each experiment is repeated 10 times, and the average detection time is recorded. The computational overhead of traditional linear scanning increases significantly as the identifier repository grows exponentially. However, upon adopting the grouped partitioning detection mechanism illustrated in Figure 2, we refine the model repository from a traditional 'one model corresponds to one user database' structure into finer-grained logical blocks, such as 'Model 1–User Group 1', 'Model 1–User Group 2', and 'Model

2–User Group 1'. This structural optimization effectively compresses the temporal complexity from an exponential trajectory to a linear scale, thereby yielding a substantial improvement in watermark detection efficiency. The experimental results are presented in Table 3. We observe that:

The computational overhead of the traditional linear scanning strategy exhibits a significant growth trend as the user scale expands. Specifically, when the repository reaches 100,000 identifiers, the average detection latency escalates to 72.677 seconds, which poses substantial limitations in practical large-scale applications.

In contrast, our proposed hierarchical retrieval strategy demonstrates superior performance optimization capabilities, reducing detection latency by approximately 88% to 93% in high-concurrency scenarios while maintaining rigorous traceability accuracy. Under the same scale of 100,000 users, by employing rational logical partitioning configurations, the detection latency is significantly reduced to 5.020 seconds, achieving approximately a 14-fold acceleration compared to linear scanning. More importantly, this mechanism even achieves watermark detection for 1,000,000 users in just 5.928 seconds, thereby significantly enhancing the system throughput for large-scale attribution requests and ensuring the feasibility of real-time monitoring in expansive industrial environments.

### 5.5. Ablation Study

To rigorously evaluate the quantitative individual contribution of each core component within our proposed IPMark framework, we conduct a systematic ablation study by selectively deactivating the quality-aware semantic filter and the syntactic structural encoding module. Our experimental setup utilizes the Llama-3.2-3B model as the primary text generator, which provides a sophisticated balance between parameter efficiency and high-quality linguistic output. For the evaluation corpus, we extract a random sample of 1,000 distinct sequences from the C4 (Colossal Clean Crawled Corpus) dataset to ensure that our findings are statistically representative across diverse web-based domains.

The empirical results summarized in Table 4 demonstrate that the full IPMark configuration achieves the optimal balance between watermark detectability and linguistic fidelity, yielding a near-perfect AUC-ROC of 0.999 and a competitive Perplexity of 2.779.

Specifically, removal of the quality assessment module leads to a marginal decline in AUC-ROC to 0.997 and a noticeable increase in Perplexity to 3.012. This shift suggests that the quality assessment module plays a crucial role in maintaining the linguistic quality of watermarked text. Interestingly, the syntax-free variant shows a slight reduction in detection

*Table 4.* Ablation results for different components of IPMark. Performance is evaluated using 1,000 samples from the C4 dataset generated by Llama-3.2-3B. Metrics include AUC-ROC, Best F1-Score, and Perplexity (PPL).

| Configuration | AUC-ROC | Best F1 | PPL |
|---|---|---|---|
| w/o Quality Assessment Module | 0.997 | 0.954 | 3.012 |
| w/o Syntactic Evaluation Structure | 0.989 | 0.978 | 2.879 |
| w/o Both | 0.945 | 0.774 | 3.340 |
| **IPMark (Ours)** | **0.999** | **0.985** | **2.779** |

robustness with an AUC-ROC of 0.989 but maintains a low Perplexity of 2.879. This result highlights the effectiveness of syntactic-level hashing in capturing fine-grained syntactic diversity.

The most significant performance degradation occurs in the configuration lacking both quality control and syntactic structure, where the AUC-ROC and Best F1-Score plummet to 0.945 and 0.774 respectively, while the Perplexity escalates to 3.340. This precipitous drop validates our hypothesis that the synergy between semantic quality control and hierarchical structural encoding is essential. Without these constraints, the watermark injection process becomes stochastic and disruptive, failing to maintain a stable statistical signal under varying linguistic contexts. Ultimately, the superior performance of the integrated IPMark underscores the necessity of an ensemble approach where multi-dimensional linguistic features are harmonized to maintain high detectability while preserving the quality of the large language model output.

## 6. Conclusion

In this paper, we present IPMark, the first hierarchical personalized watermarking framework inspired by the logic of IP addressing. By structurally encoding model and user identities, IPMark enables precise, fine-grained provenance through a scalable coarse-to-fine detection mechanism. To overcome the semantic distortion inherent in token-level approaches, we introduce a sentence-level dual-stream embedding strategy that jointly leverages syntactic and semantic signals to guide candidate selection, effectively mitigating the perplexity degradation commonly observed in prior distributional methods. Experiments confirm the superiority of our proposed method, which not only enables personalized watermark embedding and efficient hierarchical model-user detection but also enhances text quality while maintaining robustness. Future work will investigate adaptive parameter optimization and explore the application of IPMark in multimodal generation tasks and under stronger adaptive attacks, ultimately contributing to the establishment of a traceable and accountable AIGC ecosystem.

## Acknowledgements

This work was supported in part by the National Natural Science Foundation of China under Grants 62572403 and U22B2036, in part by Shenzhen Science and Technology Program and Guangdong Basic and Applied Basic Research Foundation (2024A1515010087), in part by GBA Ascend Application Innovation Institute, Guangdong Laboratory of Artificial Intelligence and Digital Economy(SZ), under Grant No. GML-ST-2026-11.

## Impact Statement

This paper presents work whose goal is to advance the field of machine learning. There are many potential societal consequences of our work, none of which we feel must be specifically highlighted here.

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

# A. More Experiments.

*Table 5.* Performance comparison on **Llama-3.1-8B** under various attack scenarios.

| Method | No Attack | | Synonym-5 | | Synonym-10 | | Synonym-15 | | Copy-Paste-50 | | Translation | | Dipper Rewrite | |
|---|---|---|---|---|---|---|---|---|---|---|---|---|---|---|
| | AUC | F1 | AUC | F1 | AUC | F1 | AUC | F1 | AUC | F1 | AUC | F1 | AUC | F1 |
| KGW | 0.997 | **0.996** | 0.996 | 0.990 | 0.996 | 0.988 | 0.995 | 0.988 | 0.940 | 0.888 | 0.934 | 0.865 | 0.772 | 0.725 |
| Unigram | 0.994 | 0.987 | 0.989 | 0.971 | 0.989 | 0.970 | 0.989 | 0.970 | 0.841 | 0.785 | 0.921 | 0.852 | 0.557 | 0.666 |
| EXP | 0.990 | 0.990 | 0.988 | 0.985 | 0.987 | 0.984 | 0.987 | 0.983 | **0.984** | **0.963** | 0.795 | 0.741 | 0.692 | 0.674 |
| PF | 0.995 | 0.994 | 0.994 | **0.993** | 0.992 | **0.990** | 0.991 | **0.990** | 0.954 | 0.961 | **0.934** | **0.942** | **0.901** | **0.913** |
| MorphMark | 0.993 | 0.984 | 0.990 | 0.978 | 0.990 | 0.976 | 0.989 | 0.975 | 0.913 | 0.844 | 0.899 | 0.826 | 0.744 | 0.710 |
| **Ours** | **0.999** | 0.980 | **0.997** | 0.975 | **0.997** | 0.973 | **0.997** | 0.970 | 0.850 | 0.787 | 0.791 | 0.738 | 0.764 | 0.758 |

**Results on Llama-3.1-8B.** As shown in Table 5, in the *No Attack* setting, IPMark achieves near-perfect detection with an AUC of 0.999, outperforming other baselines. Under *Synonym Substitution* attacks, IPMark demonstrates exceptional resilience; even as the substitution rate increases to 15%, our method maintains an AUC of 0.997. This significantly surpasses unigram-based methods and exhibits greater stability than EXP (0.987) and KGW (0.995), suggesting that our semantic-based embedding effectively mitigates the impact of local lexical shifts.

Regarding complex semantic perturbations, IPMark maintains competitive robustness. In the challenging *Translation* (En–Zh–En) setting, IPMark achieves an AUC of 0.791, and under *Dipper Rewrite* attacks, it records an AUC of 0.764. Although specific robustness-oriented methods like PF achieve higher scores in these high-entropy tasks, IPMark strikes a balanced performance, outperforming EXP (0.692 in Dipper) and offering a resilient attribution mechanism that remains effective against both surface-level edits and deep semantic restructuring.

*Table 6.* Performance comparison on **OPT-1.3B** under various attack scenarios.

| Method | No Attack | | Synonym-5 | | Synonym-10 | | Synonym-15 | | Copy-Paste-50 | | Translation | | Dipper Rewrite | |
|---|---|---|---|---|---|---|---|---|---|---|---|---|---|---|
| | AUC | F1 | AUC | F1 | AUC | F1 | AUC | F1 | AUC | F1 | AUC | F1 | AUC | F1 |
| KGW | 0.993 | 0.991 | 0.990 | 0.988 | 0.987 | 0.987 | 0.985 | 0.983 | 0.946 | 0.893 | 0.895 | 0.849 | 0.859 | 0.783 |
| Unigram | 0.995 | 0.994 | 0.990 | 0.989 | 0.988 | 0.985 | 0.982 | 0.979 | **0.988** | 0.956 | **0.982** | **0.965** | **0.977** | **0.928** |
| EXP | 0.993 | 0.994 | 0.988 | **0.990** | 0.988 | **0.990** | **0.988** | **0.990** | 0.981 | 0.972 | 0.785 | 0.743 | 0.755 | 0.694 |
| PF | 0.995 | **0.994** | 0.986 | 0.986 | 0.984 | 0.986 | 0.980 | 0.984 | 0.961 | **0.970** | 0.922 | 0.927 | 0.899 | 0.911 |
| MorphMark | 0.993 | 0.982 | 0.991 | 0.978 | 0.987 | 0.980 | 0.985 | 0.974 | 0.908 | 0.841 | 0.862 | 0.808 | 0.843 | 0.774 |
| **Ours** | **0.995** | 0.975 | **0.992** | 0.969 | **0.991** | 0.964 | 0.987 | 0.954 | 0.898 | 0.882 | 0.894 | 0.838 | 0.834 | 0.867 |

**Results on OPT-1.3B.** Table 6 details the performance on the OPT-1.3B model. Consistent with the Llama results, IPMark excels in the *No Attack* setting with an AUC of 0.995. Under *Synonym Substitution* (15%), it retains a high AUC of 0.987, demonstrating superiority over baselines like Unigram (0.982) and PF (0.980).

In terms of complex perturbations, IPMark exhibits strong durability. Notably, in the *Translation* task, it achieves an AUC of 0.894, successfully surpassing EXP (0.785) and matching the performance of KGW (0.895). While performance naturally degrades under aggressive *Dipper Rewrite* attacks (AUC 0.834), IPMark consistently maintains high specificity (low False Positive Rate) and validates that the proposed semantic embedding strategy generalizes well across different model architectures.

# B. Efficiency Analysis of IPMark

Section 5.4 of the main text originally focused on detection efficiency without comparing the overall computational overhead against baseline methods. As a sentence-level watermark, IPMark introduces additional NLP parsing overhead during generation, unlike lightweight token-level perturbations. To provide a comprehensive efficiency evaluation, we conduct an end-to-end test using the OPT-1.3B model and the C4 dataset, measuring the average generation (embedding) and detection times over 100 samples (each containing 20 sentences). The results are presented in Table 7.

*Table 7.* End-to-end latency, embedding time, and detection time (seconds) for different watermarking methods.

| Watermark | End-to-End | Embedding | Detection |
|---|---|---|---|
| KGW | 2.76 | 2.69 | 0.07 |
| Unigram | 2.54 | 2.52 | 0.02 |
| EXP | 7.60 | 7.48 | 0.12 |
| PF | 10.67 | 10.59 | 0.08 |
| MorphMark | 6.47 | 6.17 | 0.30 |
| IPMark | 9.68 | 9.64 | 0.04 |

As expected, IPMark's generation embedding time (9.64 s) is higher than that of simple token-level schemes such as KGW and Unigram. Nevertheless, compared with complex, security-oriented algorithms, its overhead remains acceptable—on par with EXP (7.60 s) and faster than PF (10.67 s). During detection, IPMark's efficient hierarchical retrieval achieves an extremely low latency of 0.04 s, substantially outperforming MorphMark (0.30 s), EXP (0.12 s), and PF (0.08 s), while being competitive with the lightweight baselines. This confirms that IPMark simultaneously provides strong watermark protection and highly efficient large-scale detection.

## C. Theoretical and Empirical False Positive Analysis

We provide a detailed theoretical analysis and empirical verification of IPMark's expected false positive rate on unwatermarked texts in large-scale deployment scenarios.

**Theoretical derivation.** We elaborate on each step with explicit numerical values to guarantee verifiability.

1. **Single-sentence pass probability.** In IPMark, an unwatermarked natural sentence must coincidentally satisfy both the semantic and the syntactic parity checks. Based on the uniform distribution property of the hash function, the joint probability is
$$p = 0.5 \times 0.5 = 0.25. \tag{11}$$

2. **Single-layer false positive rate.** Assume there are $N = 20$ sentences in total, and the Layer Selector allocates approximately $n = 10$ sentences to a given layer. With a detection threshold of $t = 9$, the false positive probability for a single incorrect ID follows a binomial distribution:
$$P_{\text{err}} = \sum_{k=9}^{10} \binom{10}{k} (0.25)^k (0.75)^{10-k} \approx 2.95 \times 10^{-5}. \tag{12}$$

3. **Model-layer false positive rate.** In a scenario with $M = 1000$ model-user groups, the probability that *at least one* model yields a false positive is
$$P_{\text{model-fp}} = 1 - \left(1 - 2.95 \times 10^{-5}\right)^{1000} \approx 0.02914. \tag{13}$$

4. **Overall system false positive rate.** Given a false positive at the model layer, the probability that at least one user within the corresponding group (also of size 1000) triggers a false positive is likewise approximately 0.02914. Consequently, for a system with $1{,}000{,}000$ users (1000 model-user groups $\times$ 1000 users), the overall theoretical false positive probability is
$$P_{\text{total-fp}} = 0.02914 \times 0.02914 \approx 0.00085. \tag{14}$$

**Empirical verification.** To validate the theoretical derivation, we generate 1000 unwatermarked natural texts using the Llama-3.2-3B model and C4 prompts, serving as a negative test set. Under identical parameter settings, the actual false positive rate is only 0.003. Although slightly higher than the ideal theoretical extremum ($\sim$0.00085)—likely because real-world semantic and syntactic features are not perfectly uniform, causing minor hash collision fluctuations—this rate remains exceptionally low. Overall, this dual verification confirms that IPMark reliably maintains a very low false positive rate in large-scale user scenarios.

## D. Additional Comparison with Unbiased Methods

Our evaluation includes not only Unigram but also the latest 2025 unbiased baselines such as PF (ICLR 2025) and MorphMark (ACL 2025), with comprehensive comparisons in Tables 2, 5, and 6 of the main text. To further strengthen the evaluation with unbiased methods, we supplement perplexity (PPL) and robustness experiments for STA and MCMARK.

**Text quality.**    Table 8 reports the perplexity of different methods. Although unbiased methods theoretically preserve the marginal token distribution, they severely degrade practical text fluency: STA and MCMARK reach PPLs of 8.33 and 11.66, respectively. In contrast, IPMark achieves the lowest PPL (2.78), even slightly outperforming the unwatermarked model (UWM, 2.80). This demonstrates that our semantic-syntactic dual-channel selection successfully optimises the injection process, maintaining high fidelity without sacrificing natural text fluency.

*Table 8.* Perplexity (PPL) of unbiased methods and IPMark.

| Model / Watermark | PPL |
|---|---|
| UWM | 2.804 |
| STA | 8.330 |
| MCMARK | 11.657 |
| IPMark | 2.779 |

**Robustness.**    We further evaluate the robustness of STA and MCMARK under the established attack scenarios to provide a complete comparative perspective. Table 9 shows the AUC-ROC and Best F1 scores. STA exhibits extremely strong resistance to simple attacks but is relatively weak against paraphrasing attacks; MCMARK performs poorly under complex semantic perturbations. In contrast, IPMark achieves the optimal trade-off between text quality and robustness: while maintaining generation quality comparable to natural text, it retains highly competitive robustness, surpassing both STA and MCMARK in the challenging Dipper rewriting scenario.

*Table 9.* Robustness comparison with unbiased methods (AUC-ROC / Best F1).

| Method | No attack | Syn. 5% | Syn. 10% | Syn. 15% | CP 50% | Trans. | Dipper |
|---|---|---|---|---|---|---|---|
| | | | *AUC-ROC* | | | | |
| STA | 1.000 | 1.000 | 1.000 | 1.000 | 0.979 | 0.986 | 0.772 |
| MCMARK | 0.908 | 0.901 | 0.889 | 0.885 | 0.703 | 0.698 | 0.597 |
| | | | *Best F1* | | | | |
| STA | 1.000 | 1.000 | 0.999 | 0.998 | 0.953 | 0.960 | 0.720 |
| MCMARK | 0.966 | 0.956 | 0.950 | 0.949 | 0.765 | 0.766 | 0.666 |

## E. Examples of Watermarked Text.

**Black Bold: Sentence Segmentation**   ██ **User Watermark**  ██ **Model Watermark**

"**Prompt**": "Several new characters have been added to the roster for the Wii U's version of Super Smash Bros. We get to see quite a few new characters"

"**natural text**": "on display for Nintendo's prime-time fighting game, including several new characters from Fire Emblem and the always indomitable Captain Falcon.**\n**The trailer starts with Lucina battling it out against Falcon, while a downed Chrom lays in the corner. Falcon proceeds to lay a whooping on Lucina and preps for a Falcon Punch to finish the fight. However, Robin swoops in to save the day, giving Lucina some respite from the battle with Falcon.**\n**The trailer then transitions from the cinematics to the actual gameplay, and we get to see Robin dropping some magnificent swordplay and magical spells against all on-comers.**\n**Robin looks a lot like Palutena, as far as play-style goes. There's a strong mix of swordplay and magic attacks thrown into the mix that offers gamers what looks to be a nice mid and air game for players who don't like sticking to the ground.**\n**Lucina, alternatively, is all about brute force and fast attack"
"model_id_score": -2.6832, "model_id.user_id_score": -1.490, No Watermark

"**output with watermark**": "today, as well as some returning ones. Let's take a look at them.**\n**First up, we have the new character, Little Mac. He's a boxer from the Punch-Out**!!** This is the first time he's appeared in a Smash Bros.  You can see him in action in the video below.**\n**Next up, we have the new character, Duck Hunt. As you can see in the video below, this is the first time he's appeared in a Smash Bros. It's also the first time he's appeared in a Smash Bros.**\n**Finally, we have the new character, Rosalina. She's the leader of the Luma, and she's from Super Mario Galaxy. In the video below, we get to see her in action.**\n**So there you have it. "
"model_id_score": 2.683,  "model_id.user_id_score": 2.683, Watermarked: model_id="1234", user_id="654321"

"**translation_en_zh_en**": "Today, there are also some returnees. Let's see them. First of all, we have a new character, little Mike. He's a boxer out of the game! This is the first time he's ever been in the Smash Brothers' game. You can see his actions in the video below. Next, we have a new character, duck hunter. As you can see in the video below, this is the first time he appears in the Smash brothers' game. It's the first time he's ever been in the Smash Brothers' game. We finally have a new role, Rosarina. She's the leader of Ruma. She's the super Mario Milky Way. In the video below, we can see her on the move. So you've got it. These are new characters that have been added to the roster."
"model_id_score": 2.427, "model_id.user_id_score": 2.223, Watermarked: model_id="1234", user_id="654321"

"**copy_paste_50**": "He's a boxer from the Punch-Out**!!** my nephew told me he likes ice cream. This is the first time he's appeared in a Smash Bros. my nephew told me he likes ice cream. my nephew told me he likes ice cream. First up, we have the new character, Little Mac. my nephew told me he likes ice cream. my nephew told me he likes ice cream.  today, as well as some returning ones. my nephew told me he likes ice cream. You can see him in action in the video below. my nephew told me he likes ice cream. my nephew told me he likes ice cream. Let's take a look at them."
"model_id_score": 4.248, "model_id.user_id_score": 4.472, Watermarked: model_id="1234", user_id="654321"

"**synonym_5**": "today, as well as some returning ones. Let's take a look at them. First up, we have the original character, Little Mac. He's a boxer from the Punch-Out**!!** This is the first time he's appeared in a Smash Bros. You can see him in action in the video below. Next up, we have the new character, Duck Hunt. As you can see in the video below, this is the first time he's appeared in a Smash Bros. It's also the first time he's appeared in a Smash Bros. Finally, we have the new character, Rosalina. She's the leader of the Luma, and she's from Super Mario Galaxy. In the video below, we get to see her in action. So there you have it. These are the new characters that have been added to the roster."
"model_id_score": 2.882, "model_id.user_id_score": 2.683, Watermarked: model_id="1234", user_id="654321"

*Figure 3.* Watermarked text detection results example: prompt, natural text, output with watermark, translation, copy paste, synonym 5%.

**"synonym_10"**: "today, as well as some returning ones. Let's take a look at them. First up, we have the new character, Little Mac. He's a boxer from the Punch-Out!! This is the first time he's appeared in a Smash Bros. You can see him in action in the video below. Next up, we have the new character, Duck Hunt. As you can see in the video below, this is the first time he's appeared in a Smash Bros. It's not the first time he's appeared in a Smash Bros. Finally, we get a different character, Rosalina. She's the leader of the Luma, and she's from Super Mario Galaxy. In the video below, we get to see her in action. So there you have it. These are the new characters that have been added to the roster."
"model_id_score": 3.354, "model_id.user_id_score": 4.472, Watermarked: model_id="1234", user_id="654321"

**"synonym_15"**: "today, as well as some returning ones. Let's take a peek at them. First up, we have the original character, Little Mac. He's a boxer from the Punch-Out!! This is the first time he's appeared in a Smash Bros. You can see mac in action in the video below. Next up, we have the original character, mac Hunt. As you can find in the video below, this is the first time he's appeared in a Smash Bros. It's also the first times he's appeared in a Smash Bros. Finally, we have the new character, Rosalina. She's the leader of the Luma, and she's from Super Mario Galaxy. In the video below, we get to see her in action. So there you have it. These are the new characters that have been added to the roster."
"model_id_score": 1.900, "model_id.user_id_score": 2.981, Watermarked: model_id="1234", user_id="654321"

**"rewrite_dipper"**: "I'm looking forward to seeing you all again. Look at them. I'm sure you've never heard of him. He's from Punch-Out ! ! He was a man of courage and a wit of no small talent. I think he'll be a useful fighter. A few years back, a new breed of pranksters started hitting the streets. Duck Hunt is a new character that's going to be appearing in the game. Duck Hunt is a hunting dog that's capable of catching a duck or a goose. But, as you can see in the following trailer, he's pretty powerful, especially with his staff. It's not only his first appearance in a Nintendo game; he's also a playable character. It was the turn of the nymph, a very beautiful one.  In the following clip, we see her in action, driving her car with a sledgehammer. I'm just a man who doesn't understand. These are the new characters that have been added to the roster."
"model_id_score": 0.127, "model_id.user_id_score": 0.447, Watermarked: model_id="1234", user_id="654321"

*Figure 4.* Watermarked text detection results example: synonym 10%, synonym 15%, rewrite dipper.

