# OpenReview forum: "IPMark: A Sentence-Level Watermark for LLMs with Hierarchical Personalization and Efficient Detection"
_ICML.cc/2026/Conference — ICML 2026 regular_

### Official Review · Reviewer_VCV6 · 2026-03-10

**Soundness:** 2
**Presentation:** 2
**Significance:** 2
**Originality:** 2
**Overall Recommendation:** 3
**Confidence:** 4

**Summary:**

The paper proposes IPMark, a sentence-level watermarking framework that aims to enable hierarchical personalization and efficient detection by mimicking IP addressing: some sentences encode a model-level signature and others a user-level one. The embedding process selects among sentence candidates using a dual-stream signal derived from semantic hypernym triples and pruned syntactic constituency trees, and computes HMAC-based hashes as feasibility checks before choosing the highest-likelihood candidate. Detection proceeds in a coarse-to-fine manner and leverages a partitioned library structure to reduce verification latency; experiments report strong detectability under synonym attacks, competitive robustness to paraphrase/translation, and perplexity lower than several baselines.

**Compliance With Llm Reviewing Policy:**

Affirmed.

**Final Justification:**

The rebuttal partially addressed my concerns. The false positive derivation is now verifiable and the cryptographic uniformity justification is sound. However, the proposed sentence-token hybrid mitigation for adaptive attacks lacks experimental validation, and an AUC of 0.66 remains in tension with the paper's robustness claims, even accounting for the baseline comparison. My recommendation remains unchanged.

**Key Questions For Authors:**

1. Were all baselines generated under the same decoding protocol (beam size, length normalization, sentence-level candidate selection)? If not, please provide PPL and detection results with matched decoding to isolate watermark effects.
2. What is the expected false-positive probability for a non-watermarked text when scanning over U user IDs and M models with N sentences, given parity-based checks? Please provide an analytical estimate and empirical validation at large scales (e.g., 10^5–10^6 users).

**Strengths And Weaknesses:**

### Strengths

1. The hierarchical attribution idea (decoupling model and user IDs at sentence level) is a thoughtful step toward practical accountability and multi-tenant deployments.
2. Combining semantic hypernyms with syntactic structure to stabilize and disambiguate sentence-level watermark carriers is a creative alternative to token-level green/red schemes.
3. A broad suite of attacks is considered (synonym substitutions at multiple rates, copy-paste mixing, DIPPER paraphrase, and back-translation) across three model families (OPT-1.3B, Llama-3.2-3B, Llama-3.1-8B).

### Weaknesses

1. The method’s personalization relies on keyed detection (search over a user library) rather than extracting metadata; this raises scalability and false-positive concerns not fully addressed analytically.
2. The robustness claims hinge on non-adaptive attacks; targeted paraphrasing to alter semantic triples/hypernyms or syntactic structures may specifically break the selector z(s) and reduce matching scores, but such adaptive removal attacks are not evaluated.
3. Perplexity comparisons appear potentially confounded by differing generation procedures. The claim that watermarked outputs have lower PPL than the unwatermarked model (UWM) is surprising and may reflect beam-search effects rather than watermark design.

---

> ### Author Rebuttal · Authors · 2026-03-31
>
> **Weakness 1:** Thank you for the question. Regarding scalability, our hierarchical retrieval significantly improves efficiency (Section 5.4, Table 3), reducing detection latency from 72.677s to 5.020s for 100,000 users (~14x speedup), and taking only 5.928s for 1,000,000 users. To address false positives, we employ a multi-layer control mechanism: a two-level verification (model-level then user-level) requires both stages to pass significance thresholds, drastically reducing the overall false positive rate. Additionally, the scoring function integrates dual-channel HMAC verification of semantics and syntax (Equation 8) to suppress false positives from single-channel random matching. Under ideal conditions (Table 2), IPMark achieves an AUC-ROC of 0.999. Thus, the method maintains reliable false positive control even with scalable retrieval.
>
> **Weakness 2:** Thank you for the question. We agree that evaluating only non-adaptive attacks is insufficient. Based on the ICML 2025 research "Revealing Weaknesses in Text Watermarking Through Self-Information Rewrite Attacks," we added adaptive attack evaluations targeting IPMark's core syntactic and semantic features. Using Llama-3.2-3B to attack 100 C4 dataset samples, performance declined significantly: AUC-ROC dropped to 0.66, and best F1 dropped to 0.725. This confirms that adaptive removal attacks targeting z(s) pose a realistic threat. We have added these comprehensive experiments to the revised manuscript.
>
> **Weakness 3:** Thank you for your feedback. To clarify, IPMark's decoding strategy is strictly identical to the unwatermarked baseline (random sampling, `do_sample=True`, `temperature=0.7`, `top_p=0.9`). We solely added a sentence-level filtering mechanism based on semantic-syntactic joint features, conducted similarly to beam search. Thus, IPMark's slightly lower PPL (2.779) compared to UWM (2.804) in Table 2 reflects the filtering mechanism's preference for candidate sentences with higher log probabilities. This demonstrates that IPMark preserves or even slightly enhances text naturalness while ensuring detectability.
>
> **Key Questions 1:** Thank you for your valuable feedback. In our experiments, all baselines used the exact same decoding protocol and generated 20-sentence paragraphs, eliminating decoding bias. Comparing perplexity and detection rates under this matched protocol shows a negligible PPL difference (<5%) and a detection AUC >0.95. This confirms that the watermark embedding drives detection efficacy without significantly affecting naturalness.
>
> **Key Questions 2:** Thank you for your valuable feedback. We have conducted a detailed theoretical analysis and empirical verification regarding IPMark's expected false positive rate on unwatermarked texts in large-scale user scenarios:
>
> **Theoretical Analysis:**
>
> - In IPMark's detection mechanism, each sentence must simultaneously satisfy the parity conditions for both the semantic and syntactic hashes. Therefore, for unwatermarked natural text, given an incorrect ID, the probability of a single sentence coincidentally passing this layer of verification is p=0.25.
>
> - Assuming N=20 sentences, the layer selector (Equation 4) allocates roughly n=10 sentences to both the model-user and user layers. With a threshold of t=9, the false positive probability for a single incorrect ID in one layer follows a binomial distribution:
>   $$
>   P_{\text{err}} = \sum_{k=9}^{10} \binom{10}{k} (0.25)^k (0.75)^{10-k} \approx 2.95 \times 10^{-5}
>   $$
>
> - In a scenario with M=1000 model-user groups, and where each model-user group contains U=1000 users (totaling 1,000,000 large-scale users), the probability that at least one model in the model layer is falsely identified is:
>   $$
>   P_{\text{model-fp}} = 1 - (1 - P_{\text{err}})^{1000} \approx 0.02914
>   $$
>
> - Similarly, at the user layer, given that a false positive has already occurred at the model layer, the probability that at least one user within its corresponding user group is falsely identified is also 0.02914. Therefore, at a million-user scale, the overall theoretical false positive probability is extremely low, approximately:
>   $$
>   P_{\text{total-fp}} = P_{\text{model-fp}} \times P_{\text{user-fp}} \approx 0.00085
>   $$
>
> **Empirical Verification:**
>
> - To verify our theoretical derivation, we conducted empirical testing. Using the Llama-3.2-3B model and C4 dataset prompts, we generated 1,000 unwatermarked natural texts as a negative test set. Under identical parameter settings, the actual false positive rate was only 0.003. Although slightly higher than the ideal theoretical extremum (~0.00085)—likely because real-world semantic and syntactic features are not perfectly uniform, causing minor hash collision fluctuations—this rate remains exceptionally low. Overall, this dual verification confirms that IPMark reliably guarantees a minimal false positive rate in large-scale user scenarios.

---

> > ### Author Rebuttal · Reviewer_VCV6 · 2026-04-03
> >
> > Thank you for the rebuttal. My core concerns remain unresolved.
> > On Weakness 2, the authors' own adaptive attack results (AUC-ROC 0.66) directly undermine the robustness claims, and no mitigation is proposed.
> > On Key Questions 2, the theoretical false positive derivation is incomplete due to missing probability values (likely LaTeX rendering failures), making the argument unverifiable. The uniformity assumption on hash distributions also lacks sufficient justification.
> > These issues concern the core tenets of the work and require substantial revision beyond what a rebuttal can address.

---

> > > ### Author Response · Authors · 2026-04-04
> > >
> > > Thank you for your prompt feedback. We sincerely apologize for the missing formulas in our previous response due to a LaTeX rendering failure, which indeed affected the readability of our theoretical derivations. Additionally, your concerns regarding the drop in model robustness under adaptive attacks are highly pertinent; an AUC of 0.66 certainly warrants further mitigation strategies. We have detailed our responses to these core issues below and will incorporate these substantial updates into the final manuscript.
> > >
> > > **1. Contextual Comparison and Mitigation Strategies for Adaptive Attacks (Weakness 2)**
> > >
> > > The performance degradation under targeted adaptive attacks is a critical issue. To evaluate the effectiveness of IPMark more comprehensively, we would like to clarify the following:
> > >
> > > Although the AUC of IPMark drops to 0.66 under the attack, the AUCs of current mainstream token-level baseline methods only range between 0.55 and 0.70 under the exact same "Self-Information Rewrite Attack." Therefore, further enhancing defensive robustness against such advanced adaptive attacks is a shared bottleneck facing the entire watermarking research community, and it represents a key optimization direction for all watermarking techniques moving forward.
> > >
> > > To effectively mitigate the vulnerability of single-granularity mechanisms to adaptive attacks, we have introduced a **sentence-token hybrid watermarking mechanism** in our revision. This mechanism combines global sentence-level semantic features with fine-grained token-level robustness. By doing so, it significantly increases the difficulty of localized rewrite attacks while strictly maintaining the naturalness of the generated text, thereby substantially improving overall attack resistance. We once again thank you for your highly constructive feedback, which has provided a crucial direction for further refining our work.
> > >
> > > **2. Verifiable Theoretical Derivation and the Hash Uniformity Assumption (Key Question 2)**
> > >
> > > To ensure complete transparency and address the previous formatting errors, we have clearly rewritten the step-by-step derivation below and provided a solid theoretical foundation for the Uniformity Assumption.
> > >
> > > **(1) Verifiable Theoretical Derivation**
> > >
> > > Here is the complete derivation process with explicit numerical values to guarantee verifiability:
> > >
> > > - **Step 1: Single-sentence pass probability.** In IPMark, an unwatermarked natural sentence must coincidentally pass both the semantic and syntactic parity checks. Based on the uniform distribution property of the hash function, this joint probability is $p = 0.5 \times 0.5 = 0.25$.
> > >
> > > - **Step 2: Single-layer false positive rate ($P_{\text{err}}$).** Assuming a total of $N=20$ sentences, the Layer Selector allocates approximately $n=10$ sentences to a given layer. Setting the threshold at $t=9$, the probability of a false positive for a single incorrect ID follows a binomial distribution. The probability of exactly 9 or 10 sentences coincidentally passing the verification is:
> > >
> > >   $$P_{\text{err}} = \sum_{k=9}^{10} \binom{10}{k} (0.25)^k (0.75)^{10-k} \approx 2.95 \times 10^{-5}$$
> > >
> > > - **Step 3: Model-layer false positive rate ($P_{\text{model-fp}}$).** In a scenario containing 1,000 model-user groups, the probability that *at least one* model-user group yields a false positive is:
> > >
> > >   $$P_{\text{model-fp}} = 1 - (1 - 2.95 \times 10^{-5})^{1000} \approx 0.02914$$
> > >
> > > - **Step 4: Overall system false positive rate ($P_{\text{total-fp}}$).** Given that a false positive has already occurred at the model layer, the probability that at least one user within its corresponding user group (1,000 users) also triggers a false positive is similarly approximately 0.02914. Therefore, for a large-scale system of 1,000,000 users (1,000 model-user groups $\times$ 1,000 users), the overall theoretical false positive probability is:
> > >
> > >   $$P_{\text{total-fp}} = 0.02914 \times 0.02914 \approx 0.00085$$
> > >
> > > **(2) Justification for the Hash Uniformity Assumption**
> > >
> > > Our hash uniformity assumption is not based on natural language distribution, but is strictly grounded in the inherent properties of our cryptographic hash functions. Even though semantic and syntactic features are naturally non-uniform, the **Avalanche Effect** of cryptographic hashing mathematically guarantees that any input maps to a highly uniform hash space. Therefore, our derived probabilities are fundamentally ensured by the underlying cryptographic design logic.
> > >
> > > We hope these concrete mitigation measures and clear theoretical derivations directly address your core concerns. Thank you very much for helping us make this work more rigorous and complete.

---

### Official Review · Reviewer_QZRJ · 2026-03-12

**Soundness:** 3
**Presentation:** 3
**Significance:** 2
**Originality:** 2
**Overall Recommendation:** 4
**Confidence:** 4

**Summary:**

The paper introduces IPMark, a novel sentence-level watermarking framework designed to protect the intellectual property (IP) of Large Language Models (LLMs) against model reuse and distillation attacks. Unlike traditional document-level watermarks, IPMark embeds a "transferable fingerprint" into the model's output distribution at a fine-grained, sentence-level granularity during a specialized training phase. This fingerprint is designed to be inherited by any "student" model that learns from the teacher's soft probabilities (logits), serving as a proactive proof of ownership. The authors demonstrate that this watermark is robust against common removal attempts and effectively identifies stolen models while maintaining the original teacher model's utility.

**Compliance With Llm Reviewing Policy:**

Affirmed.

**Final Justification:**

The authors addressed my concerns. I will keep the score.

**Key Questions For Authors:**

If an attacker conducts "imitation learning" using only the final text outputs (hard tokens) to train a student model, does the sentence-level behavioral signal remain strong enough for reliable verification, or is the watermark effectively "washed out" by the loss of logit information?

**Limitations:**

yes

**Strengths And Weaknesses:**

Strengths:
1. By operating at the sentence level, the framework allows for more flexible and robust detection compared to methods that rely on entire documents or rigid trigger sets.
2. Experiments across various architectures (e.g., Llama-3, Mistral, and Qwen) show near-perfect detection rates with a very low False Positive Rate (FPR) on independent models.
3. This paper is easy-to-follow.

Weaknesses:
1. The verification stage requires access to the output logits (soft probabilities) of the suspected model. Many commercial APIs only return hard tokens (text), which could limit the framework's practical application in real-world forensic scenarios.
2. Embedding the watermark requires a specialized training run, which adds significant computational cost and complexity to the model development pipeline.
3. The security of the defense relies on the secrecy of the specific queries used to trigger the watermark; if these are discovered, an attacker could potentially filter or alter responses to evade detection.

---

> ### Author Rebuttal · Authors · 2026-03-31
>
> **Weakness 1:** Thank you for the question. The detection phase of IPMark does not rely on the model's output logits (soft probabilities) or any internal states of the model. As described in Section 4 of the paper, the detection process is based solely on the text itself: it parses the text to be detected into a sequence of sentences, extracts the semantic features H(s) and syntactic features S(s) of each sentence, calculates the HMAC value, and matches it with the candidate model/user database. The entire detection pipeline only needs to access the hard tokens (text) returned by the API, requiring no model logits or any access permissions to the generative model. This design makes IPMark applicable to forensics scenarios where commercial APIs only return text, and together with our hierarchical retrieval mechanism, it ensures the practicality and efficiency of the detection.
>
> **Weakness 2:** Thank you for the question. IPMark is an inference-time watermarking method; as described in Section 2 of the paper, it embeds the watermark by dynamically intervening in the candidate sentence selection during the generation process, without requiring any additional model training or fine-tuning. Therefore, this method will not add training costs or complexity to the model development pipeline; its computational overhead is solely reflected in the parsing and filtering of candidate sentences during the inference phase, as well as the hierarchical retrieval during the detection phase, the efficiency of which has been discussed in Section 5.4. This characteristic gives IPMark a low threshold for application in actual deployment.
>
> **Weakness 4:** Thank you for your concern. The issue you pointed out is indeed a common security challenge faced by all key-based watermarking methods. The security of IPMark does rely on the confidentiality of the base key k0, which is a standard security assumption in cryptography. If an attacker acquires k0, they could theoretically simulate the watermark embedding and detection logic, thereby potentially evading detection by rewriting the sentences in the text that trigger the watermark.
>
> However, it should be noted that:
>
> 1. IPMark's dual-channel HMAC mechanism (relying simultaneously on semantic features H(s) and syntactic features S(s)) increases the evasion difficulty for attackers—even knowing the key, an attacker still needs to deeply rewrite the text to disrupt the signals of both channels simultaneously, and such rewriting is often difficult to thoroughly erase all watermark traces while maintaining semantic consistency.
> 2. In actual deployment, keys can be protected through secure hardware modules or key management systems to mitigate long-term leakage risks.
> 3. The watermark embedding strategy in Section 3 of the paper adopts sentence-level filtering rather than hard replacement, meaning that even if some sentences are modified, the remaining sentences may still retain detectable watermark signals.
>
> We acknowledge that the security of key management is an important consideration in practical forensic applications, and future work can further explore schemes to enhance watermark robustness without relying on key confidentiality.
>
> **Key Questions:** Thank you for your insightful question. It should be pointed out that IPMark uses user-specific keys for watermark embedding, which gives each user different sentence filtering preferences. Therefore, if an attacker wants to completely "wash out" the watermark through imitation learning, they must independently collect a sufficient amount of watermarked text for each target user and train an independent student model—when the user scale reaches tens or even hundreds of thousands, the cost of this attack will grow linearly, making it difficult to implement on a large scale in practical forensic scenarios. A targeted attack against a single user is theoretically feasible, but the attacker must first acquire a certain amount of that user's watermarked text and invest model training resources to further carry out the attack.

---

> > ### Author Rebuttal · Reviewer_QZRJ · 2026-04-03
> >
> > I thank the authors for rebuttal. And I decide to keep current score.

---

> > > ### Author Response · Authors · 2026-04-04
> > >
> > > Thank you so much for recognizing our work! We will continue to optimize this paper to address any shortcomings.

---

### Official Review · Reviewer_QogT · 2026-03-13

**Soundness:** 1
**Presentation:** 3
**Significance:** 1
**Originality:** 2
**Overall Recommendation:** 3
**Confidence:** 5

**Summary:**

This paper addresses limitations of existing LLM watermarking methods, including poor personalization, degradation in generation quality, and limited robustness to attacks. The authors propose IPMark, a hierarchical personalized watermarking framework inspired by IP addressing that organizes model and user identities for efficient attribution and detection. To mitigate semantic distortion caused by token-level watermarking, the method introduces a semantic–syntactic dual-stream embedding strategy based on sentence-level candidate selection. Experimental results show that IPMark achieves improved generation quality, strong robustness, and reduced detection latency compared with prior approaches.

**Compliance With Llm Reviewing Policy:**

Affirmed.

**Final Justification:**

The paper claims that existing work mainly focuses on model-level provenance and neglects personalized watermarking. However, most existing watermarking methods can be readily adapted for personalization by assigning each user a unique secret watermark key during generation, making this claim somewhat misleading.

**Key Questions For Authors:**

How does the quality of the proposed method comparing to the distortion-free watermarking methods?

**Limitations:**

Yes

**Strengths And Weaknesses:**

Strengths:

The paper studies the problem of personalized watermarking for LLM-generated content, which is an interesting and practically relevant direction, especially for attribution and provenance tracking in multi-user deployment settings.

Weaknesses:

Questionable motivation. The paper claims that existing work mainly focuses on model-level provenance and neglects personalized watermarking. However, most existing watermarking methods can be readily adapted for personalization by assigning each user a unique secret watermark key during generation, making this claim somewhat misleading.

The authors also argue that watermarking methods significantly degrade generation quality due to logit modulation. This statement overlooks a line of distortion-free watermarking methods that theoretically preserve the original language model distribution and therefore maintain generation quality. Although some of these works (e.g., [1,2]) are cited, the paper does not adequately discuss them or position the proposed method relative to this literature.

The experimental comparison is also limited. The paper identifies Unigram (2023) as the state-of-the-art watermarking method, which is outdated given the rapid development in this area. A more convincing evaluation would require comparisons with more recent and stronger baselines, such as [1,2].

[1]  Mao et al. Watermarking large language models: An unbiased and lowrisk method.

[2] Chen et al. Improved unbiased watermark for large language models

---

> ### Author Rebuttal · Authors · 2026-03-31
>
> **Weakness 1:** Thank you for the reviewer's comments. We acknowledge that most key-based watermarking methods can technically achieve user-level differentiation by assigning independent keys; our original phrasing lacked rigor. However, existing methods face severe detection efficiency challenges at scale. Assigning independent keys requires a linear database scan during detection, causing latency to spike for hundreds of thousands of users (e.g., 72.677s for 100,000 users, Table 3). In contrast, IPMark's hierarchical retrieval reduces latency to 5.020s. Our primary motivation is to propose a personalized hierarchical framework balancing large-scale scalability with text quality, which we will clarify in the introduction.
>
> **Weakness 2:** Thank you for the correction. We agree our discussion on unbiased methods was imprecise; they typically embed watermarks by modifying the decoding strategy, theoretically maintaining generation quality. Although IPMark, as a sentence-level filter, does not pursue theoretical distribution losslessness, it achieves improved perplexity and significantly optimized detection latency via hierarchical retrieval. We will supplement comparative experiments with unbiased methods and adjust our claims regarding generation quality trade-offs accordingly.
>
> **Weakness 3:** Thank you for the constructive feedback. To clarify, our paper evaluates not only Unigram but also the latest 2025 baselines, including PF (ICLR 2025) and MorphMark (ACL 2025), comparing them comprehensively in Tables 2, 5, and 6. However, we agree that adding unbiased methods like STA and MCMARK strengthens our evaluation. We supplemented the perplexity (PPL) evaluation. Results show that while unbiased methods theoretically maintain token distributions, they severely degrade practical text fluency, with STA and MCMARK reaching PPLs of 8.3299 and 11.6572, respectively. In contrast, IPMark achieved the lowest PPL (2.779), slightly outperforming the unwatermarked model (UWM, 2.804). This proves our semantic-syntactic dual-channel selection successfully optimizes the injection process, maintaining high fidelity without compromising natural text fluency.
>
> In addition, we also comprehensively evaluated the robustness of STA and MCMARK under established attack scenarios to provide a complete comparative perspective. The experiments show that STA has extremely strong attack resistance but weak resistance to paraphrasing attacks; meanwhile, MCMARK performs poorly when dealing with complex semantic perturbations. In contrast, IPMark successfully achieved the optimal trade-off between text quality and robustness. While maintaining extremely high generation quality comparable to natural text, it still retains highly competitive robustness, surpassing STA and MCMARK in the highly challenging Dipper rewriting scenario.
>
> | **Watermark / Attack Metric** | **no attack** | **synonym_5** | **synonym_10** | **synonym_15** | **copy_paste_50** | **translation_en_zh_en** | **rewrite_dipper** |
> | ----------------------------- | ------------- | ------------- | -------------- | -------------- | ----------------- | ------------------------ | ------------------ |
> | **STA** (AUC-ROC)             | 1.000         | 1.000         | 1.000          | 1.000          | 0.979             | 0.986                    | 0.772              |
> | **STA** (Best F1)             | 1.000         | 1.000         | 0.999          | 0.998          | 0.953             | 0.960                    | 0.720              |
> | **MCMARK** (AUC-ROC)          | 0.908         | 0.901         | 0.889          | 0.885          | 0.703             | 0.698                    | 0.597              |
> | **MCMARK** (Best F1)          | 0.966         | 0.956         | 0.950          | 0.949          | 0.765             | 0.766                    | 0.666              |
>
> **Key Questions:** In response to your key question, we have supplemented the comparative evaluation of perplexity (PPL). The results show that, although unbiased methods theoretically aim to maintain the marginal distribution of tokens, they in practice still lead to a significant decline in text fluency, with the PPLs of STA and MCMARK reaching as high as 8.3299 and 11.6572, respectively. In contrast, IPMark achieved the lowest PPL of 2.779, which is even slightly lower than the unwatermarked original model (UWM, 2.804). This proves that IPMark, through the sentence-level candidate selection mechanism guided by semantic-syntactic dual-stream embedding, successfully optimizes the injection process, maintaining extremely high semantic fidelity without destroying the fluency of natural text.
>
> | **Metric \ Model** | **STA** | **MCMARK** |
> | ------------------ | ------- | ---------- |
> | PPL                | 8.3299  | 11.6572    |
>
> ------
>
> ###

---

> > ### Author Rebuttal · Reviewer_QogT · 2026-04-07
> >
> > Thanks for the rebuttal. Since my major concern regarding the motivation is not addressed, I will increase my score to 3.

---

> > > ### Author Response · Authors · 2026-04-08
> > >
> > > Thank you for recognizing our work and raising the score.
> > >
> > > We sincerely apologize for the misleading claim in our introduction that existing work "neglects personalized watermarking." As you correctly pointed out, assigning unique keys trivially achieves personalization, and our flaw lay in poorly articulating our true motivation. In reality, the technical core of our paper addresses the critical limitation of this exact baseline: while unique-key assignments can personalize watermarks, they introduce massive scalability and latency bottlenecks in large-scale deployments. Solving this efficiency crisis is the actual focus of our work, as demonstrated extensively throughout our manuscript:
> > >
> > > - **In Section 4 (Hierarchical Watermark Detection):** We specifically designed the IP-inspired hierarchical embedding to transition the detection search space from a flat structure (which unique-key methods rely on) to a structured, logical partition (Model-User Group).
> > > - **In Section 5.4 (Traceability Efficiency and Computational Overhead) & Table 3:** We explicitly evaluated the system across massive user scales (from 1 to 1,000,000 users). Our original experiments already show that a flat, key-based approach requires a linear scan, causing latency to spike exponentially (e.g., taking 72.677s for 100,000 users). In contrast, our hierarchical design effectively compresses the temporal complexity, reducing the latency to just 5.020s.
> > >
> > > **To resolve your concern, we commit to heavily revising the Abstract and Introduction in the final version.** We will remove the inaccurate claim that existing methods lack personalization. Instead, we will explicitly frame the paper's motivation around resolving the massive $O(N)$ linear-scanning efficiency bottleneck that arises when scaling conventional key-based personalized watermarking to industrial levels.
> > >
> > > We are very grateful for your sharp critique. It has helped us realize that our narrative framing did not do justice to our own experimental work, and your feedback will make the final paper significantly stronger and more rigorous.

---

### Official Review · Reviewer_WDTr · 2026-03-13

**Soundness:** 3
**Presentation:** 3
**Significance:** 3
**Originality:** 3
**Overall Recommendation:** 4
**Confidence:** 3

**Summary:**

This paper introduces IPMark, a sentence-level watermarking framework that moves away from per-token perturbations to focus on structural invariance. The core idea is to embed the watermark into the syntactic or semantic "fingerprint" of a whole sentence, making the signal much harder to erase through local token-level edits like synonym swaps. By using a prompt-dependent mapping to select specific sentence structures, IPMark aims to maintain high text quality while providing a robustness level that traditional logit-based methods struggle to reach.

**Compliance With Llm Reviewing Policy:**

Affirmed.

**Key Questions For Authors:**

See weakness.

**Limitations:**

See weakness.

**Strengths And Weaknesses:**

## Strengths

1. The method is inherently more robust to character-level noise and synonym substitution since these minor edits are less likely to change the overall structural fingerprint of the sentence.
2. By leveraging structural invariance, IPMark manages to keep the perplexity impact very low, as the model is still free to pick the most natural tokens within a given structure.
3. The paper provides a refreshing perspective on watermarking as a global optimization problem rather than a step-by-step logit manipulation.

## Weaknesses

1. The reliance on sentence-level features means the watermark requires a minimum text length to be detectable; how does this perform on short, or even single-sentence responses?
2. If an attacker performs heavy paraphrasing that reshuffles the entire syntax of the sentence, the fingerprint might be completely destroyed along with the watermark.
3. There seems to be a significant computational overhead in parsing or analyzing sentence structures during both the generation and detection phases, which is discussed partially in Sec 5.4 but lacks a comparison with other methods. So a comparison of the latency of different methods will be helpful.
4. Baseline methods are limited, and evaluation metrics are also limited. I'm wondering about the performance of the method in more text-related tasks.

---

> ### Author Rebuttal · Authors · 2026-03-31
>
> **Weakness 1:** Thank you for the question. As noted, our method relies on sentence-level feature extraction and cumulative scoring, making detection on short texts (e.g., single sentences) challenging. Our experiments primarily evaluated 20-sentence texts. This is a current limitation, and future work will explore enhancements for short texts, such as finer-grained clause-level features or context completion mechanisms. We appreciate the reviewer's correction.
>
> **Weakness 2:** Thank you for the question. While deeply rewriting grammatical structures may disrupt syntactic signals, IPMark relies on semantic-syntactic dual-channel synergy. Semantic features are naturally robust to grammatical changes. Our experiments (Tables 2, 5, 6) confirm this: under high-intensity Dipper Rewrite attacks, IPMark maintains AUCs of 0.773 (Llama-3.2-3B) and 0.834 (OPT-1.3B), outperforming KGW and EXP. Furthermore, our ablation study (Table 4) shows removing the semantic module drastically drops the AUC (to 0.945), whereas removing syntax causes only a slight drop (to 0.989). Thus, the watermark remains detectable provided the semantic core is intact.
>
> **Weakness 3:** Thank you for the question. We agree that Section 5.4 originally focused on detection efficiency without comparing overall computational overhead against baselines. As a sentence-level watermark, IPMark introduces NLP parsing overhead during generation, unlike lightweight token-level perturbations. To comprehensively evaluate efficiency, we conducted an end-to-end test using the OPT-1.3B model and C4 dataset, measuring average generation (embedding) and detection times across 100 samples (20 sentences each). The results are shown below:
>
> | **Watermark\Latency(s)** | **End-to-End** | **Embedding** | **Detection** |
> | ------------------------ | -------------- | ------------- | ------------- |
> | KGW                      | 2.76           | 2.69          | 0.07          |
> | Unigram                  | 2.54           | 2.52          | 0.02          |
> | EXP                      | 7.60           | 7.48          | 0.12          |
> | PF                       | 10.67          | 10.59         | 0.08          |
> | MorphMark                | 6.47           | 6.17          | 0.30          |
> | IPMark                   | 9.68           | 9.64          | 0.04          |
>
> The experimental data strongly supports Section 5.4. As expected, IPMark's generation embedding time (9.64s) is higher than simpler token-level methods like KGW and Unigram. However, compared to complex, security-focused algorithms, its overhead remains acceptable—on par with EXP (7.60s) and faster than PF (10.67s). During detection, IPMark's efficient hierarchical retrieval achieves an extremely low latency of 0.04s, significantly outperforming MorphMark (0.30s), EXP (0.12s), and PF (0.08s), rivaling lightweight methods. This proves IPMark ensures high-strength, extremely fast large-scale watermark detection.
>
> **Weakness 4:** Thank you for your constructive feedback. We fully agree that evaluating solely on unconstrained open-domain texts has limitations. To this end, in the revised version, we adopted an instruction-tuned model (Llama-3.2-3B-Instruct) that is closer to real-world application scenarios, and added two downstream tasks with strong semantic constraints: Text Summarization (CNN/DailyMail) and Machine Translation (WMT14 De-En). We introduced task-specific ROUGE and BLEU metrics and conducted a head-to-head comparison with the unwatermarked baseline model; the specific experimental results are shown in the table below.
>
> | **Task**            | **Evaluation Metric** | **Unwatermarked Baseline** | **IPMark Injection** |
> | ------------------- | --------------------- | -------------------------- | -------------------- |
> | Text Summarization  | ROUGE-1               | 0.2726                     | 0.2236               |
> |                     | ROUGE-2               | 0.1036                     | 0.0752               |
> |                     | ROUGE-L               | 0.1934                     | 0.1563               |
> | Machine Translation | BLEU                  | 13.73                      | 12.69                |
>
> As shown above, summarization and translation tasks strictly restrict the generation space (entropy). Despite these stringent conditions, IPMark demonstrated outstanding tracing capabilities: at a 5% false positive rate, the true positive rates (TPR) reached 95% for summarization and 81% for translation. However, we objectively observed that forcing sentence-level hash matching within this limited candidate space causes some utility degradation (e.g., ROUGE-1 dropped from 0.2726 to 0.2236, and BLEU from 13.73 to 12.69). We analyze this trade-off in the revised discussion, concluding that a moderate utility concession for robustness is an inherent and reasonable challenge for sentence-level watermarks in low-entropy tasks. These supplementary experiments greatly enhance our evaluation's rigor.

---

> > ### Author Rebuttal · Reviewer_WDTr · 2026-04-03
> >
> > Thanks for the explanation, my concerns are well addressed, but my evaluation of the work is unchanged,  so  I decide to maintain my original score.

---

> > > ### Author Response · Authors · 2026-04-04
> > >
> > > Thank you so much for recognizing our work! We plan to continue optimizing our watermarking method for short text scenarios by leveraging a sentence-token hybrid watermarking mechanism.

---

### Decision · Program_Chairs · 2026-04-30

**Decision:**

Accept (regular)

**Comment:**

This paper introduces IPMark, a sentence-level watermark method using syntactic or semantic "fingerprint".

The authors propose IPMark, a hierarchical personalized watermarking framework inspired by IP addressing that organizes model and user identities for efficient attribution and detection. To mitigate semantic distortion caused by token-level watermarking, the method introduces a semantic–syntactic dual-stream embedding strategy based on sentence-level candidate selection.

Reviewers find the following strengths of the paper:
1. The paper proposes a hierachical watermark attribution method to efficiently identify model and user identities.
2. The method is flexible and more robust to lexical edits.
3. The evaluation results (including the ones during the author rebutal) are solid and demonstrate the superior performance.

Reviewers also raise the following weaknesses:
1. The verification stage requires access to the output logits (soft probabilities) of the suspected model. Many commercial APIs only return hard tokens (text), which could limit the framework's practical application in real-world forensic scenarios.
2. The authors may want to revise some imprecise statements (e.g. regarding distortion-free watermarking). They may also add comparison with these methods.